# Prospective Multicentric Assessment of ^68^Ga-DOTANOC PET/CT in Grade 1-2 GEP-NET

**DOI:** 10.3390/cancers15020513

**Published:** 2023-01-14

**Authors:** Alexandre Lugat, Éric Frampas, Yann Touchefeu, Éric Mirallié, Maëlle Le Bras, Hélène Senellart, Aurore Rauscher, Vincent Fleury, Loïc Campion, Vincent Rohmer, Olivier-François Couturier, Rachida Lebtahi, François Rouzet, Philippe Ruszniewski, Françoise Kraeber-Bodéré, Mickaël Bourgeois, Catherine Ansquer

**Affiliations:** 1Medical Oncology Department, CHU Nantes, 44000 Nantes, France; 2Center for Research in Cancerology and Immunology Nantes-Angers (CRCINA), University of Nantes, INSERM UMR 1232, 44000 Nantes, France; 3CHU Nantes, Nantes Université, Médecine Nucléaire, 44000 Nantes, France; 4Central Department of Radiology and Medical Imaging, CHU Nantes, 44000 Nantes, France; 5The Enteric Nervous System in Gut and Brain Disorders, Université de Nantes, INSERM, TENS, IMAD, 44000 Nantes, France; 6Chirurgie Cancérologique, Digestive et Endocrinienne, Institut des Maladies de l’Appareil Digestif, CHU de Nantes, 44000 Nantes, France; 7Department of Endocrinology, Diabetology and Nutrition, L’institut du Thorax, CHU Nantes, 44000 Nantes, France; 8Department of Medical Oncology, ICO Cancer Center, 44000 Nantes, France; 9Pharmacy, ICO Cancer Center, 44000 Nantes, France; 10Department of Nuclear Medicine, ICO Cancer Center, 44000 Nantes, France; 11Biometrics, ICO Cancer Center, 44000 Nantes, France; 12Endocrinology Department, University Hospital, 49100 Angers, France; 13Nuclear Medicine Department, University Hospital, 49100 Angers, France; 14Department of Nuclear Medicine, Beaujon Hospital, 92110, Clichy, France; 15Nuclear Medicine Department, Hôpital Bichat-Claude Bernard, AP-HP, 75018 Paris, France; 16Université de Paris and Inserm U1148, 75018 Paris, France; 17Université de Paris, Department of Gastroenterology-Pancreatology, ENETS Centre of Excellence, Beaujon University Hospital (APHP), 92110 Clichy, France; 18Université de Paris, Centre of Research on Inflammation, INSERM U1149, 75018 Paris, France; 19Groupement d’Intérêt Public Arronax, 44800 Saint-Herblain, France

**Keywords:** neuroendocrine tumors, ^68^Ga-DOTANOC, somatostatin receptor scintigraphy, CT scan, MRI

## Abstract

**Simple Summary:**

Determining the most sensitive imaging technique to evaluate neuroendocrine tumors spread may have an impact on therapeutic management. The aim of this multicentric study was to prospectively assess ^68^Ga-DOTANOC PET/CT sensitivity compared to the combination of multiphasic CT, somatostatin receptor scintigraphy and MRI to evaluate whether this imaging modality results in therapeutic modifications. We confirm in a homogenous population of 105 grade 1 or 2 gastroenteropancreatic neuroendocrine tumors the higher sensitivity of ^68^Ga-DOTANOC PET/CT in per-patient and per-region analysis, as well as in the detection of primary tumor and small lesions such as peritoneal carcinomatosis and bone lesions leading to an impact on therapeutic management of almost half of the patients.

**Abstract:**

The aim of this multicentric study was to prospectively compare ^68^Ga-DOTANOC PET/CT versus somatostatin receptor scintigraphy (SRS) with SPECT/CT, combined with multiphasic CT scan and MRI in patients with grade 1 or 2 gastroenteropancreatic neuroendocrine tumors (GEP-NET). Patients with histologically proven grade 1 or 2 GEP-NET with suspicion of recurrence or progression, or with typical aspects of GEP-NET on morphological imaging, were explored with conventional imaging (CI): SRS with SPECT/CT, multiphasic CT scan and/or liver MRI followed by ^68^Ga-DOTANOC PET/CT. The gold standard was based on histology and imaging follow-up. The data of 105 patients (45 woman and 60 men; median age) were analyzed. ^68^Ga-DOTANOC PET/CT sensitivity was significantly higher than CI sensitivity in per-patient (98.9% vs. 88.6%, *p* = 0.016) and per-region (97.6% vs. 75.6%, *p* < 0.001) analyses, in the detection of the primary (97.9% vs. 78.7%; *p* = 0.016), peritoneal carcinomatosis (95% vs. 30%, *p* < 0.001), and bone metastases (100% vs. 33.3%, *p* = 0.041). ^68^Ga-DOTANOC PET/CT had an impact on the therapeutic management of 41.9% (44/105) patients compared to decisions based on CI explorations. Our data confirm the superiority of ^68^Ga-DOTANOC PET/CT over CI in the detection of peritoneal carcinomatosis and bone metastasis, as well as its strong therapeutic impact on the management of patients with grade 1-2 GEP-NETs.

## 1. Introduction

Neuroendocrine tumors (NETs) are a heterogenous group of well differentiated neuroendocrine neoplasms (NEN) arising from the diffuse neuroendocrine system. These cells originate from the ectoderm and have the ability to produce hormones and peptides [1]. Given this whole-body distribution, NETs have been described nearly everywhere, but gastroenteropancreatic neuroendocrine tumors (GEP-NETs) are the most common. NETs are said to be rare, but taken all together their prevalence is higher than gastric or pancreatic adenocarcinomas. Moreover, their incidence is increasing, with a 6.4-fold increase from 1973 to 2012 mainly explained by an earlier diagnosis and the incidental diagnosis of non-symptomatic tumors through improved imaging techniques [2]. One of the main characteristic of NETs is the overexpression of somatostatin receptors (SSTR), especially type 2 (SSTR2) [3,4]. This overexpression has permitted the development of a phenotypic molecular imaging using somatostatin analogues [5]. Somatostatin receptor scintigraphy (SRS) using ^111^In-diethylenetriamine-pentaacetic acid-octreotide (Octreoscan^®^) was widely used since the 1980s for routine diagnostic and follow-up for patients with NETs [6]. In the past, SRS had been reported to be more sensitive than morphological imaging, with proven clinical impact in the management of patients with GEP-NETs [7,8].

In the last decade, positron emission tomography (PET), combined with computed tomography (CT) using ^68^Ga-labeled somatostatin analogues (SSA) has been developed, showing better diagnostic performances than SRS [9,10]. There are three different ^68^Ga-labeled SSA, with a DOTA as chelator: DOTATOC, DOTANOC and DOTATATE. The major difference among these radiotracers relies on a different affinity to SSTR subtypes [11], but no clinically significant differences have been demonstrated in terms of lesion detection [12,13].

As expected, ^68^Ga-labeled SSA PET/CT demonstrated higher diagnostic performances compared with SRS even optimized with single photon emission computed tomography combined with CT (SPECT/CT), leading many centers to substitute ^68^Ga-labeled SSA PET/CT to SRS for the diagnosis, detection of primary site, staging and follow up of patients with NETs [9,10,14]. Nevertheless, despite the clinical evidence of the interest of this modality, the published series are frequently heterogeneous in terms of patient populations: prospective and comparative studies are lacking in literature [15,16,17,18,19,20]. Further data are needed to confirm the interest of ^68^Ga-labeled SSA PET/CT over conventional imaging (CI), including SRS with SPECT/CT.

Moreover, better diagnostic performance does not necessarily translate into a change in therapeutic management. Only few studies tend to show this, but this change is still debated, as most of the studies were retrospective or included several types or grades of NETs [17,18,21,22,23].

The objectives of this prospective study were to compare the patient and regional-based performances of ^68^Ga-DOTANOC PET/CT and CI including SRS with SPECT/CT in grade 1 or 2 GEP-NET patients, and to determine whether performing a ^68^Ga-DOTANOC PET/CT has an impact on the therapeutic management in comparison with CI.

## 2. Materials and Methods

### 2.1. Patients and Study Design

The patients were prospectively enrolled in the Nuclear Medicine departments of Nantes, Angers and Beaujon (France) University Hospital Centers, and in the Nuclear Medicine department of Cancer Center of Saint-Herblain between November 2012 and May 2015.

Inclusion criteria were: previous or current histological proven grade 1-2 GEP-NET according to WHO 2010 classification; typical aspects of GEP-NET on morphological imaging associated or not with clinical or biological signs of GEP-NET; suspicion of recurrence or progression of GEP-NET on morphological imaging or on laboratory tests; and clinical and biological syndromes strongly suggestive of GEP-NET without identification of lesion on morphological imaging. These clinical and biological syndromes included but were not limited to carcinoid syndrome, organic hypoglycemia, Zollinger-Ellison Syndrome, and ACTH-dependent hypercortisolism. Both functional and non-functional GEP-NETs were included.

Exclusion criteria were multiple endocrine neoplasia syndrome and other genetic predisposition syndromes, non-grade 1-2 GEP-NETs, pregnancy or lactation, persons protected by law, restlessness or inability to lie still for at least 1 h, claustrophobia, other progressive cancers (except basal cell carcinoma and cancer in situ of the cervix), kidney failure that does not permit injection of contrast agent, treatment with radiotherapy, chemotherapy or other antitumor treatment within 6 weeks of previous morphological and scintigraphy examinations.

The study was approved by the Ethics Committee of Nantes University Hospital, the French authorities (the ANSM: Agence nationale de sécurité du médicament et des produits de santé and the ASN: Autorité de Sûreté Nucléaire) and was registered on clinicaltrials.gov (NCT01747096). A written informed consent was obtained in accordance with provisions of the Declaration of Helsinki for each patient.

Clinical and biological data were collected and CI including SRS were performed in a two-month period prior to the ^68^Ga-DOTANOC PET/CT. Pathology data were completed before or after the PET/CT when it was performed to characterize a lesion. The tumor grade was redefined for the final analysis according to the World Health Organization (WHO) 2019 [24].

### 2.2. Conventional Imaging

The CI included a 4 phase Multidetector CT of the chest, abdomen, and pelvis (nonenhanced images followed by arterial, portal and late phase after injection of iodinated intravenous contrast material), whole body SRS performed at 4 h and 24 h after intra-venous injection of about 122 MBq of 111In-diethylenetriamine-pentaacetic acid-octreotide (Mallinckrodt Pharmaceuticals, Dublin, Ireland) with a thoraco-abdominal-pelvic SPECT/CT at 24 h post-injection. When necessary, especially in case of a pancreas NET or liver metastasis, a liver magnetic resonance imaging (MRI) was performed.

### 2.3. Rabiolabelling and PET/CT Acquisition

DOTANOC (50 µg full GMP peptide; Iason GmbH) was radiolabelled with Gallium-68 using a fully automated synthesis method and a prepurification of eluate on a SCX cartridge [25], with Modular-Lab PharmTracer (Eckert & Ziegler, Berlin, Germany). The generators used were the Obninsk^®^ from Iason and the GMP grade IGG101 generator from Eckert & Ziegler Radiopharma Gmb.

PET/CT images were acquired in all centers according to standardized protocols, 60 min after intravenous injection of about 150 MBq (5 to 10 µg) of ^68^Ga-DOTANOC without any preparation in supine position. CT without contrast agent was performed first, then was immediately followed by the PET acquisition from vertex to mid-thighs. Patients on long-acting SSA were asked, if possible, to have their last injection 4 weeks before the imaging. None of the included patients were treated with short-acting SSA.

### 2.4. Safety Monitoring

Adverse events (AEs) were assessed using the Common Terminology Criteria for Adverse Event Classification version 3.0. The assignment of the causality for every AE was made by the investigator responsible for the care of the participant. Any AEs post injection were recorded and followed until resolution. Vital signs and clinical tolerance were assessed before and during the 2 h following injection of the radiopharmaceutical.

### 2.5. Imaging Analysis

In each center, images were not blinded analyzed by nuclear medicine physicians with an experience in NETs. In case of discrepancy, a third reviewer was asked. Focal uptake was considered to be pathological if it did not correspond to the physiological uptake of ^68^Ga-DOTANOC (as pituitary gland, spleen, adrenals and urinary tract).

Gold standard was defined on anatomopathological examination. When pathological examination was not possible for technical or ethical reasons, lesions were defined by functional and/or morphological imaging with a follow-up of at least 12 months.

A lesion was considered to be positive in CI if it was detected by at least one imaging method among whole body SRS, multi-phase CT or MRI.

A lesion detected by one of the imaging methods was considered as true-positive (TP) when confirmed by another imaging method (including other neuroendocrine tracer in cases of lesion only detected by somatostatin imaging) and follow-up or histopathology. A false-positive lesion (FP) was defined as a finding on an imaging method that was not confirmed by other imaging methods, by histopathology or follow-up. A negative finding on an imaging method was considered as false-negative (FN) if positive by histopathology or at least one other imaging method, and follow-up and as a true negative (TN) if confirmed negative by histopathological or imaging methods and follow-up. For the per-regional analysis, we defined primary tumor, lymph node, liver, peritoneal carcinomatosis, bone and other sites as regions.

### 2.6. Clinical Impact Assessment

To evaluate prospectively the therapeutic impact of ^68^Ga-DOTANOC PET/CT, we sent two questionnaires by patient to the referring physician. First, we asked what treatment was planned after performing CI. Second, once ^68^Ga-DOTANOC PET/CT was performed, we asked the physician if PET/CT results had led to a change in the planned treatment. We considered an intermodal change when the initial planned treatment was excluded in favor of another modality (such as surgery to active surveillance, or chemotherapy to surgery, or change of systemic treatment type between chemotherapy, SSA or targeted therapy). We considered an intramodal change when the initially planned treatment was modified such as a more/less extended surgery, switching from one systemic treatment to another, or a modification in timing or modality of active surveillance. Therapeutic and staging modifications after performing ^68^Ga-DOTANOC PET/CT were independently collected and validated by an expert committee (local multidisciplinary team and/or expert in GEP-NETs).

### 2.7. Statistical Analysis

Numerical data were assembled and then analyzed with Microsoft Excel 2019 v.16.46. Statistical analyses were performed with GraphPad Prism 8 v8.4.3 software. Results are presented as means or medians with standard deviation for continuous variables and as numbers and percentages for categorical variables.

The sensitivity (Se), specificity (Sp), positive predictive value (PPV), negative predictive value (NPV) and accuracy of ^68^Ga-DOTANOC PET/CT, SRS alone and CI were calculated at patients’ and regions’ levels taking into account primary, locoregional lymph nodes and metastatic regions (liver, bone, peritoneal carcinomatosis, and other sites). Because of the paired design, the McNemar test was used to compare the performance of imaging methods. The test was considered statically significant when *p* value < 0.05.

## 3. Results

### 3.1. Patients Included

One hundred and thirty patients were enrolled on the study between November 2012 and May 2015, from which 25 patients were excluded from the study: 7 of them due to a time between CI and PET/CT > 3 months, 7 because of non-compliance with imaging procedures, 6 because of a lack of clinical data or response to the impact form, 2 because of introduction of a treatment between CI and PET/CT, 1 because of a lack of gold standard (no pathological confirmation of a unique lesion detected by only one imaging method) and 1 patient withdrew his consent. One patient with a G3 NET confirmed after inclusion was also excluded.

Consequently, the data of 105 patients were finally analyzed: 60 men, 45 women, median age: 62.9-year-old (range: 29–83). The most common clinical indication of ^68^Ga-DOTANOC PET/CT was suspicion of recurrence or progression (*n* = 47), then initial staging (*n* = 41) and characterization of a lesion with typical aspect of GEP-NET on morphological exams (*n* = 17). Among them, 71 patients (67.6%) underwent a liver MRI.

The primary tumor was mostly localized, at inclusion, in the pancreas (*n* = 46), in the jejunum or ileum (*n* = 30). The primary tumor site was undetermined in 15 patients. Fifty-seven patients (54.3%) were metastatic at inclusion and 21 patients (20%) had secreting NETs.

At the end of investigations, 102 patients had NETs, 55 grade 2 and 42 grade 1. The grade was undetermined in 5 patients. The diagnosis of NET was excluded in 3 patients in whom complementary explorations revealed a follicular lymphoma, a pancreatic serous cystadenoma, and an accessory spleen. The complete clinical and demographical characteristics of patients are available Table 1.

### 3.2. Treatments Prior to Inclusion

Fifty patients (47.6%) had a first line treatment had a first line treatment prior inclusion. Among them, 35 had a surgery of the primary tumors before inclusion, including one patient who received neo-adjuvant chemotherapy. Twenty-four (22.9%) had a second line treatment, 9 patients (8.6%) had a third line treatment and 3 (2.9%) a fourth line. Details of treatments prior inclusion are available in Table 2.

### 3.3. Per-Patient Analysis

According to the gold standard, 88 patients had at least one confirmed NET lesion (primary and/or metastases).

^68^Ga-DOTANOC PET/CT was TP in 87/88 (98.9%) patients while CI was TP in 78/88 (88.6%) patients (*p* = 0.016) and SRS alone in 63/88 (71.6%) patients (*p* < 0.001). The only FN of ^68^Ga-DOTANOC PET/CT was a patient with a gastric localized G1 NET suspected by CT and confirmed by upper gastrointestinal endoscopy with biopsy. Seventeen patients had no lesion according to the gold standard. ^68^Ga-DOTANOC PET/CT was TN in 15/17 (88.2%) patients while SRS in 15/17 (88.2%) patients and CI in only 10/17 (58.8%) patients (as CI included CT and SRS results and the 5 additional FP lesions of CT). There were 2 FP of ^68^Ga-DOTANOC PET/CT also FP in SRS, CT and MRI. These two patients had suspected pancreatic lesions; the first one had one had a history of splenectomy, and the focal high uptake of ^68^Ga-DOTANOC of the tail of the pancreas in the splenic bed in his context suggested the possibility of an accessory spleen, later confirmed by splenic scintigraphy. The second went through a caudal pancreatectomy, and pathological analysis showed a serous pancreatic cystadenoma.

The overall Se, Sp, PPV, NPV and accuracy are detailed in Table 3.

### 3.4. Per-Regional Analysis

Of the 169 confirmed involved regions, 165 (97.6%) were detected by ^68^Ga-DOTANOC PET/CT and 128 (75.6%) by CI (*p* < 0.001). Only 91/169 (53.8%) were detected by SRS alone (*p* < 0.001). The mean number of regions involved per patient was 1.6 ± 1.2 with ^68^Ga-DOTANOC PET/CT, 1.4 ± 0.9 with CI and 0.9 ± 0.9 with SRS. SRS was never the most accurate imaging method for the detection of lesions and never showed any additional TP lesions compare to ^68^Ga-DOTANOC PET/CT.

^68^Ga-DOTANOC PET/CT detected primary tumor in 46/47 (97.9%) patients, CI in 37 (78.7%) patients (*p* = 0.016) and SRS in 25 (53.2%) patients (*p* < 0.001).

In 15 patients, the primary tumor site was occult before PET/CT. ^68^Ga-DOTANOC PET/CT detected the primary tumor site in 9 of them (60%): 4 in the pancreas, 3 in the jejunum or ileum and 2 in the duodenum. Figure 1A shows an example of a patient with a duodenal gastrinoma detected by ^68^Ga-DOTANOC PET/CT, later confirmed by pathological examination.

Additionally, ^68^Ga-DOTANOC PET/CT allowed to rectify the primary tumor site in one patient, confirming an ileal NET instead of an initially suspected pancreatic NET. 

Therefore, taken together, ^68^Ga-DOTANOC PET/CT modified the diagnostic localization of the primary tumor in 11 (10.5%) patients.

^68^Ga-DOTANOC PET/CT detected locoregional lymph node metastases in 38/39 (97.4%) patients while CI detected lymph node metastases in 23 (59.0%) patients (*p* < 0.001) and SRS in 17 (43.6%) patients (*p* < 0.001).

The Se, Sp, PPV, NPV and accuracy are detailed according to primary tumor and lymph node, are available in Table 4.

A total of 56 patients had one or more proven distant metastatic sites with a total of 84 different regions involved. The most common distant metastatic site was the liver (48 patients), followed by peritoneal carcinomatosis (20 patients) then bone (9 patients) and other metastatic sites including distant lymph node metastases (7 patients). ^68^Ga-DOTANOC PET/CT significantly detected a higher number of metastatic sites compared to CI: 82 (97.6%) metastatic sites vs. 64 (76.2%) (*p* < 0.001) and SRS: 46 (54.8%) (*p* < 0.001). Among the suspected metastatic sites, there was one ^68^Ga-DOTANOC PET/CT FP corresponding to a suspicion of peritoneal carcinomatosis, not confirmed by pathological analysis or follow-up.

^68^Ga-DOTANOC PET/CT detected liver involvement in 47/48 (97.9%) patients, CI in 46 (95.8%) patients (*p* = 1, NS) and SRS in only 37 patients (77.1%). ^68^Ga-DOTANOC PET/CT had a better specificity for liver metastasis than CI (*p* = 0.041; 100% with 0 FP vs. 89.5% with 6 FP, respectively).

^68^Ga-DOTANOC PET/CT was clearly the most sensitive method for carcinomatosis detection, with a per patient sensitivity of 95.0% (19/20) vs. 30.0% (6/20) for CI (*p* < 0.001) and 15.0% (3/20) for SRS (*p* < 0.001). ^68^Ga-DOTANOC PET/CT was also the most sensitive for bone metastasis detection, with a sensitivity of 100% (9/9) vs. 33.3% (3/9) for CI (*p* = 0.041). Figure 1B,C shows, respectively, bone metastasis and peritoneal carcinomatosis in patients who were not known to have metastasis in these sites, later confirmed by follow-up.

The Se, Sp, PPV, NPV and accuracy according to the metastatic site involved are available in Table 5.

### 3.5. Impact of ^68^Ga-DOTANOC PET/CT on Therapeutic Management

The discovery or modification of the suspected primary tumor site by ^68^Ga-DOTANOC PET/CT had a therapeutic consequence in 7/11 (63.6%) patients, with an intermodality modification in 6 of them and an intramodality modification in one. 

In 22 (21%) patients, ^68^Ga-DOTANOC PET/CT modified the staging of the disease. In 18 patients it was an upstaging: 7 patients became N+ and/or M+. In 4 patients it was a downstaging, with a suspicion of metastases infirmed by ^68^Ga-DOTANOC PET/CT. In 11 patients who were already M+, ^68^Ga-DOTANOC PET/CT revealed new metastatic sites. These modifications have led to therapeutic modifications in 11 patients (50%), mostly in patients who were not previously known as metastatic, with an intermodality modification in 7 of them and an intramodality modification in 4.

Taken all together, ^68^Ga-DOTANOC PET/CT had a therapeutic impact in 44 out of 105 (41.9%) patients. In 14 patients, it was an intramodality modification: the planned surgical treatment was modified by ^68^Ga-DOTANOC PET/CT in 9 patients, with a more invasive procedure in 7 patients, and a less invasive approach in 2 patients. In the other 5 patients, it was a modification in the surveillance frequency and modality.

The other 30 patients had an intermodality modification. ^68^Ga-DOTANOC PET/CT prevented unnecessary surgery in 11 patients (due to the detection of a locally more extensive disease or the detection of additional involved organs). Meanwhile, it allowed the surgery in 8 patients by the discovery of the primary or by ruling out metastasis. Overall, ^68^Ga-DOTANOC PET/CT analysis led to begin a systemic therapy in 12 patients (11.4%), which did not include peptide receptor radionuclide therapy (PRRT), and which was not available in France at the time of the study.

Details of therapeutics modifications are available in Table 6.

### 3.6. Safety Analysis

During the study, only 2 patients experienced 4 non-serious AE, considered by the data safety monitoring board as “possibly related” to the study drug administration. One patient experienced parosmia and weakness that resolved within minutes, and the other patient experienced injection site pain and arterial hypertension without clinical sign or ECG modification but with another episode the next day, which resolved without any treatment.

## 4. Discussion

Despite generally being described as a tumor with an indolent behavior, 30 to 50% of patients with GEP-NET are already metastatic at diagnosis [26,27].

Many studies have already shown the superiority of ^68^Ga-labeled somatostatin analogues PET/CT in terms of additional lesion detection compared to SRS and anatomic imaging [9,16,28,29]. To our best knowledge, our study is the first to compare diagnostic performances of a ^68^Ga-labeled SSA, and the combination of morphological imaging and SRS with SPECT/CT in a homogenous series of grade 1 and 2 GEP-NETs prospectively included.

In a per-patient analysis, ^68^Ga-DOTANOC PET/CT sensitivity and specificity were 98.9% (CI 95% 93.8–99.9) and 82.4% (CI 95% 59.0–93.8), respectively, similar to those reported in previous metanalyses: Yang et al. showed a sensitivity of 93% (CI 95% 89–96%) and a specificity of 85% (CI 95% 74–93) with ^68^Ga-DOTATOC, and a sensitivity of 96% (CI 95% 91–99%) and a specificity of 100% (CI 95% 82–100) with ^68^Ga-DOTATATE. In another study, Graham et al. reported a sensitivity of 92% (0.85–0.96) and a specificity of 82% (CI 95% 69–90) with ^68^Ga-DOTATOC [12,30]. We confirmed the superiority of ^68^Ga-DOTANOC PET/CT over SRS (Se: 71.6% and Sp: 88.2%, *p* = 0.016) but also over the combination of morphological imaging and SRS with SPECT/CT of thorax, abdomen, and pelvis (Se: 88.6%, Sp: 58.8%).

In addition to better performance in terms of per-patient analysis, our study confirmed better performances in a per-region analysis, especially in the detection of peritoneal carcinomatosis and bone metastasis. Performances of ^68^Ga-labeled SSA in the detection of bone metastases have been explored by Ambrosini et al. who reported a sensitivity and a specificity of 100%, with better performances than CT [31]. We confirmed these great performances especially in the detection of small size lesions over CI (Se: 33.3%, *p* = 0.041), probably due to better spatial resolution of PET/CT and better detection of bone marrow lesions which are sometimes difficult to detect by CI, particularly CT. Performances of ^68^Ga-labeled SSA concerning peritoneal carcinomatosis lesions are poorly described in literature. In a specific study, Norlén et al., were interested in the preoperative detection of peritoneal carcinomatosis of small intestinal NET [32]. They showed a higher sensitivity of ^68^Ga-DOTATOC and ^68^Ga-DOTATATE in a per-lesion analysis (47.5%) vs. only 12.2% with CT. In our study, we were not able to perform a per-lesion analysis due to the high number of lesions and the lack of gold standard for all the lesions detected (as all of our patients were not candidates for surgery), but at a per-region level, we also clearly showed the higher performances of ^68^Ga-DOTANOC PET/CT over SRS alone or combined with morphological imaging with a sensitivity of 95.0% (CI 95% 76.4–99.7) vs. only 15.0% and 30.0%, respectively. Despite this, carcinomatosis are often very small foci difficult to detect by imaging, and its spread remains underestimated by ^68^Ga-labeled SSA PET/CT when compared to the surgical exploration of peritoneal cavity with histopathological analysis. It is illustrated by the relatively high rate of FN lesions of the series of Norlén et al., and this should be kept in mind before undergoing surgery in NET patients.

We showed that ^68^Ga-DOTANOC PET/CT was a specific imaging with only 2 FP, especially in primary tumors detection: a pancreatic serous cystadenoma and an accessory spleen, which are common and known causes of FP with ^68^Ga-labeled SSA PET/CT [33,34,35].

In our series, 15 patients explored had occult primary after CI and SRS. ^68^Ga-DOTANOC PET/CT notably permitted the reduction in the total number of occult tumors from 15 to 9 (−60%). Primary tumors were mainly located in the pancreas, then the small intestine and duodenum. This result is particularly interesting considering the fact that NETs with unknown primary tumor have a worse prognosis, and that the therapeutic management can be different depending on the primary. This is especially the case between pancreatic and midgut NETs, mainly due to their well-established differences of chemosensitivity [27]. In particular, it is well established that primary NET could be difficult to detect because of its small size or its location in hollow or moving organs. Pruthi et al., [18] had retrospectively included 68 patients with pathologically documented metastatic NETs over 3 years. ^68^Ga-DOTANOC PET/CT identified primary sites in 40 out of these 68 patients i.e., 58.8% of cases. Interestingly, primary sites were mostly in the small intestine, followed by rectum, pancreas, and stomach. Prasad et al. shown similar results with ^68^Ga-DOTANOC PET/CT, detecting primary tumor sites in 59% of the cases mainly located in the small intestine, pancreas, rectum and colon primary [20].

Taken all together, ^68^Ga-DOTANOC PET/CT finally had a therapeutic impact in 44 out of 105 (41.9%) patients compared to decisions based on CI explorations alone. Previous studies evaluating the therapeutic impact of ^68^Ga-labeled SSA PET/CT had reported a change in clinical management ranging from 16.0 to 59.6% [19,28,29,36,37,38,39,40,41,42]. The main difference in our study is that at the period of recruitment, we did not have routine access to PRRT with ^177^Lu-DOTATATE since the marketing authorization of LUTATHERA^®^ was obtained in France only in 2018. This therapy, based on a somatostatin analogue (DOTATATE) radiolabeled with the β- emitter ^177^Lu, has deeply modified the management of ileal NETs since the results of NETTER-1 trial publication in 2017 [43], making this treatment as a second option for progressive midgut NETs in case of sufficient SSTR expression documented by SRS or ^68^Ga-labeled SSA PET/CT. It is of interest in the treatment of other primary NETs overexpressing SSTR, and is also recognized by expert committees despite the lack of randomized studies. In this context, the therapeutic impact of ^68^Ga-labeled SSA PET/CT is probably underestimated in our study.

One of the strengths of our study is its prospective design, but also its homogeneity as it was only focused on grade 1-2 GEP-NETs. Indeed, the other prospective studies interested in GEP-NETs included various grade of NETs, even grade 3 [29,39]. Moreover, most studies interested in the therapeutic impact and diagnosis performances of ^68^Ga-labeled SSA PET/CT on NETs included various primary tumors [19,28,36,37,38,39,40,41,44]. Although they share a common origin, the therapeutic implications can be quite different depending on the primary tumor site. Moreover, we excluded genetic predisposition syndromes, knowing that therapeutic management is different in these patients who may have multiple NETs in the same organ, such as pancreatic NETs in MEN1 patients.

Given the variety of treatment modalities available in GEP-NETs, ranging from active surveillance to chemotherapy, it is especially crucial to optimally characterize the disease in order to select the most appropriate therapy at initial staging, restaging or follow up. Furthermore, despite the undeniable theranostic role of ^68^Ga-labeled SSA PET/CT before PRRT, our prospective study demonstrated that beyond this choice, ^68^Ga-labeled SSA PET/CT have a strong therapeutic impact in patients with GEP-NETs at various stages of the disease over CI, including SRS with SPECT/CT.

## 5. Conclusions

Our data confirm the superiority of ^68^Ga-DOTANOC PET/CT over morphological imaging even combined with SRS, in particular in the detection of occult primary, peritoneal carcinomatosis and bone metastasis, as well as its strong therapeutic impact in the management of patients with grade 1-2 GEP-NETs. Performing this imaging modified the therapeutic management of almost half of the patients, even in patients already known to be metastatic and in the absence of available PRRT.

## Figures and Tables

**Figure 1 cancers-15-00513-f001:**
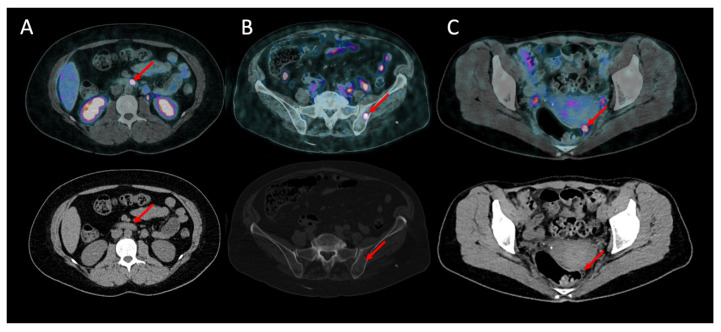
Additional lesions detected by 68Ga-DOTANOC (upper: fusion pET and CT images; lower: CT scan). (**A**) Primary lesion: grade 2 duodenal gastrinoma in a 35-year-old female. (**B**) Bone metastasis in a 50-year-old female with a grade 1 intestinal NET. (**C**) Peritoneal carcinomatosis in a 79-year-old female with a grade 2 intestinal NET.

**Table 1 cancers-15-00513-t001:** Patients’ characteristics at inclusion.

Characteristics	Value	%	*n*
Age (y; median and range)	62.8 (29–83)		105
Sex ratio (F/M)	45/60		-
Indication for PET/CT			-
Initial staging	41	39.0	-
Suspicion of recurrence or progression	47	44.8	-
Suspicion of GEP-NET on morphological exams	17	16.2	-
Secretion			-
None	84	80.0	-
Carcinoid syndrome	9	8.6	-
Insulin	5	4.8	-
Gastrin	6	5.7	-
Glucagon	1	1.0	-
NET Tumoral grade (WHO 2019)			102
1	42	41.2	-
2	55	53.9	-
Unknown	5	4.9	-
Primary tumor site confirmed or suspected at inclusion			105
Pancreas	46	43.8	-
Jejunum and ileum	30	28.6	-
Rectum	4	3.8	-
Stomach	4	3.8	-
Duodenum	4	3.8	-
Appendix	1	1.0	-
Colon	1	1.0	
Occult	15	14.3	-
Localized GEP-NET at inclusion	33	31.4	
Locoregional lymph node involvement without visceral metastases GEP-NET (N+M0) at inclusion	15	14.3	-
Distant metastatic GEP-NET (N0 or N+M+) at inclusion	57	54.3	-

GEP: gastroenteropancreatic; NET: neuroendocrine tumor; PET/CT: positron emission tomography combined with computed tomography; WHO: World Health Organization.

**Table 2 cancers-15-00513-t002:** Treatments prior to inclusion.

	First Line	Second Line	Third Line	Fourth Line
*n*	50	24	9	3
Surgery of the primary tumor	34	1		
Surgery (other than the primary tumor)	4	4		
SSA	4	10	5	1
Active surveillance	4	4	1	
(Chemo)embolization of the liver	1	1	1	
Chemotherapy	3	2	2	1
Thermoablation		2		1

SSA: Somatostatin analogues.

**Table 3 cancers-15-00513-t003:** Sensitivities, specificities, positive predictive values, negative predictive values and accuracies of SRS, Conventional Imaging and ^68^Ga-DOTANOC PET/CT at Patients’ levels.

Performance Values	SRS (%)	CI including SRS (%)	^68^Ga-DOTANOC PET/CT (%)	p PET/CT vs. CI	p PET/CT vs. SRS
Se	71.6% (63/88) [61.4–80.0]	88.6% (78/88) [80.3–93.7]	98.9% (87/88) [93.8–99.9]	*p* = 0.016	*p* < 0.001
Sp	88.2% (15/17) [55.7–97.9]	58.8% (10/17) [36.0–78.4]	88.2% (15/17) [65.7–97.9]	NS	NS
PPV	96.9% (63/65) [89.5–99.5]	91.8% (78/85) [84.0–95.6]	97.8% (87/89) [92.1–97.9]		
NPV	37.5% (15/40) [24.2–53.0]	50.0% (10/20) [29.9–70.1]	88.2% (14/15) [70.2–99.7]		
Accuracy	74.3% (78/105)	83.8% (88/105)	97.1% (102/105)		

CI: conventional imaging; NS: not significant; NPV: negative predictive value; PET/CT: positron emission tomography combined with computed tomography; PPV: positive predictive value; Se: sensitivity; Sp: specificity; SRS: somatostatin receptor scintigraphy.

**Table 4 cancers-15-00513-t004:** Sensitivities, specificities, positive predictive values, negative predictive values and accuracies of SRS, Conventional Imaging and ^68^Ga-DOTANOC PET/CT at primary tumor and lymph nodes’ levels.

Performance Values	SRS (%)	CI including SRS (%)	^68^Ga-DOTANOC PET/CT (%)	p PET/CT vs. CI	p PET/CT vs. SRS
Primary tumor					
Se	53.2% (25/47) [39.2–66.7]	78.7% (37/47) [65.1–88.0]	97.9% (46/47) [88.9–99.9]	*p* = 0.016	*p* < 0.001
Sp	81.8% (9/11) [52.3–96.8]	63.6% (7/11) [39.1–86.2]	81.8% (9/11) [52.3–96.8]	NS	NS
PPV	92.6% (25/27) [76.6–98.7]	90.2% (37/41) [77.5–96.1]	95.8% (46/48) [86.0–99.3]		
NPV	29.0% (9/31) [16.1–46.6]	41.2% (7/17) [24.6–66.3]	90.0% (9/10) [59.6–99.5]		
Accuracy	58.6% (34/58)	75.9% (44/58)	94.8% (55/58)		
Lymph node					
Se	43.6% (17/39) [29.3–59.0]	59.0% (23/39) [43.4–72.9]	97.4% (38/39) [86.8–99.9]	*p* < 0.001	*p* < 0.001
Sp	100% (66/66) [94.5–100]	92.4% (61/66) [83.5–96.7]	98.9% (65/66) [91.9–99.9]	NS	NS
PPV	100% (17/17) [81.6–100]	82.1% (23/28) [64.4–92.1]	97.4% (38/39) [86.8–99.9]		
NPV	75.0% (66/88) [65.0–82.9]	79.2% (61/77) [68.8–86.8]	98.5% (65/66) [91.9–99.9]		
Accuracy	79.1% (83/105)	80.0% (84/105)	98.1% (103/105)		

CI: conventional imaging; NS: not significant; NPV: negative predictive value; PET/CT: positron emission tomography combined with computed tomography; PPV: positive predictive value; Se: sensitivity; Sp: specificity; SRS: somatostatin receptor scintigraphy.

**Table 5 cancers-15-00513-t005:** Sensitivities, specificities, positive predictive values, negative predictive values, and accuracies of Optimized Octreoscan^®^ scintigraphy, Conventional Imaging and ^68^Ga-DOTANOC PET/CT at metastatic sites’ levels.

Performance Values	SRS (%)	CI including SRS (%)	^68^Ga-DOTANOC PET/CT (%)	p PET/CT vs. CI	p PET/CT vs. SRS
Liver					
Se	77.1% (37/48) [63.5–86.7]	95.8% (46/48) [86.0–99.3]	97.9% (47/48) [89.1–1.0]	NS	*p* = 0.004
Sp	100% (57/57) [93.7–100]	89.5% (51/57) [78.9–95.1]	100% (57/57) [93.7–100]	*p* = 0.004	NS
PPV	100% (37/37) [90.6–100]	88.5% (46/52) [77.0–95.0]	100% (47/47) [92.4–100]		
NPV	83.8% (57/68) [73.3–90.7]	96.2% (51/53) [87.3–99.3]	98.3% (57/58) [90.9–99.9]		
Accuracy	89.5% (94/105)	92.4% (97/105)	99.1% (104/105)		
Peritoneal carcinomatosis					
Se	15.0% (3/20) [5.2–36.0]	30.0% (6/20) [14.6–51.9]	95.0% (19/20) [76.4–99.7]	*p* < 0.001	*p* < 0.001
Sp	97.7% (83/85) [91.8–99.6]	97.7% (83/85) [91.8–99.9]	98.8% (84/85) [93.6–99.9]	NS	NS
PPV	60% (3/5) [23.1–92.9]	75,0% (6/8) [40.9–95.6]	95.0% (19/20) [76.4–99.7]		
NPV	83.0% (83/100) [74.5–89.1]	85.6% (83/97) [77.2–91.2]	98.9% (84/85) [93.6–99.9]		
Accuracy	81.0% (86/105)	84.8% (89/105)	98.1% (103/105)		
Bone					
Se	33.3% (3/9) [12.1–64.6]	33.3% (3/9) [12.1–64.6]	100% (9/9) [70.1–100]	*p* = 0. 041	*p* = 0.041
Sp	100% (96/96) [96.2–100]	99.0% (95/96) [94.3–100]	100% (96/96) [96.2–100]	NS	NS
PPV	100% (3/3) [43.9–100]	75.0% (3/4) [30.1–98.7]	100% (9/9) [70.1–100]		
NPV	94.1% (96/102) [87.8–97.3]	94.1% (95/101) [87.6–97.3]	100% (96/96) [96.2–100]		
Accuracy	94.3 (99/105)	93.3 (98/105)	100% (105/105)		
Other sites					
Se	42.9% (3/7) [15.8–75.0]	71.4% (5/7) [35.9–94.9]	100% (7/7) [64.6–100]	NS	NS
Sp	98.0 % (96/98) [92.9–99.6]	94.9% (93/98) [88.6–97.8]	100% (98/98) [96.2–100]	NS	NS
PPV	60.0 % (3/5) [23.1–93.0]	50.0% (5/10) [23.7–76.3]	100% (7/7) [64.6–100]		
NPV	96.0 % (96/100) [90.2–98.4]	97.9% (93/95) [92.7–99.6]	100% (98/98) [96.2–100]		
Accuracy	94.3 % (101/105)	93.3% (100/105)	100% (105/105)		

CI: conventional imaging; NS: not significant; NPV: negative predictive value; PET/CT: positron emission tomography combined with computed tomography; PPV: positive predictive value; Se: sensitivity; Sp: specificity; SRS: somatostatin receptor scintigraphy.

**Table 6 cancers-15-00513-t006:** Impact of 68Ga-DOTANOC on the clinical management.

Management Modifications after ^68^Ga-DOTANOC PET/CT	Frequency	%
Intramodality	14/105	13.3
More extensive surgical procedure	7	6.7
Extension of surgery to the lymph nodes	3	2.9
Extension of primary tumor resection	2	1.9
Liver metastasis resection	1	1.0
Less extensive surgical procedure	2	1.9
Cancelation of liver resection	1	1.0
Cancelation of lymph node resection	1	1.0
Modification in surveillance	5	4.8
Intermodality	30/105	28.6
Surgery indication	8	7.6
Surgery deferred	11	10.5
Initiation of SSA	7	6.7
Chemotherapy indication	2	1.9
Targeted therapy indication	3	2.9
Chemoembolization of the liver indication	1	1.0
Surveillance indication	7	6.7
Discontinuation of specific surveillance	1	1.0

## Data Availability

Data available on request due to restrictions eg privacy or ethical.

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
