# Peer review of "Prospective Multicentric Assessment of 68Ga-DOTANOC PET/CT in Grade 1-2 GEP-NET"

_cancers, 2023, doi:10.3390/cancers15020513_

Round 1
Reviewer 1 Report
Dear colleagues,
I would like in the first place to congratulate with the Authors for the amount and the quality of the work done.
The omogeneity of the patients and the protocol used seems really solid, and the multicentric and prospectic model helps with the soundness of the results.
I would like to appoint:
-in “2. Materials and Methods”:
2.1: Would be useful to clearly state also here that both functioning and non-functioning GEP-NETs were included; I would also highlight that would be useful to better define “clinical and biological syndrome strongly suggestive of GEP-NET…”; I would also like to ask what the “complete … biological data” consisted of and whether the samples were centrally analyzed or not.
2.2-2.3: I would like to ask whether both the CI and the PET-CT where centrally or peripherally analyzed and in case both were evaluated if the central radiological review was blinded or not, if there were any cases of divergence between the two, and how you managed it in that cases.
-in Table.1: It would be useful to have a little bit broad amount of patients characteristics (ie weight, BMI etc) and of biological data, hence those have been collected.
Reviewer 2 Report
The manuscript compares 68Ga-DOTANOC PET/CT to conventional imaging for grade 1 or 2 gastroenteropancreatic neuroendocrine tumor patients in per-patient and per-region analyses. This paper is particularly interesting to readers because of the low volume of prospectively collected data for patients undergoing 68Ga-DOTANOC PET/CT. Additionally, the inclusion of the investigation into the impact of PET/CT on therapeutic management is valuable.
Specific recommendations for revision are below:
The term primary tumor should be used together, not simply as “primary.”
The long sentences in section 2.6 are unclear and should be reworded to better describe how the clinical impact was assessed. It becomes clear when the results are read, but the initial description should be improved.
The description of treatments prior to inclusion seems incomplete. For patients on second-, third-, or fourth-line therapy, what did they have before? I understand if this information was not collected, but then this should be explained and the timing of when the treatments “prior to inclusion” listed in table 2 should be explained as well.
A short explanation in section 3.3 as to why there are fewer true negatives in the combined CI group compared to the SRS only group could be interesting or helpful for the readers.
For the per-region analysis, please describe what the defined regions are.
There are English language errors throughout the paper. The writing is understandable, but there are quite a few of these minor errors. For example, many times the wrong preposition is used. For example, “DOTANOC had an impact in therapeutic management;” on should be used instead of in. The verb tenses should be checked and there are a few word choices that are used incorrectly (e.g. conducting/conducted is used incorrectly twice). There are also some typos throughout; for example, 80s does not need an apostrophe.
Some abbreviations have not been defined, for example, NEN, TN, NS.
The appearance of Table 1 could be improved by using left indent and without bullet points.
The bottom of Figure 1 is not visible on the PDF.
